# Energy Management System for Polygeneration Microgrids, Including Battery Degradation and Curtailment Costs

**DOI:** 10.3390/s24227122

**Published:** 2024-11-05

**Authors:** Yassine Ennassiri, Miguel de-Simón-Martín, Stefano Bracco, Michela Robba

**Affiliations:** 1Department of Computer Science, Bioengineering, Robotics, and Systems Engineering (DIBRIS), University of Genoa, Via Opera Pia 13, 16145 Genoa, Italy; yassine.ennassiri@edu.unige.it; 2Department of Electrical Engineering, Systems, and Automation, Universidad de León (University of León), Campus de Vegazana s/n, 24071 León, Spain; miguel.simon@unileon.es; 3Department of Electrical, Electronics, and Telecommunication Engineering and Naval Architecture, University of Genoa, Via Opera Pia 11a, 16145 Genoa, Italy; stefano.bracco@unige.it

**Keywords:** energy management system, energy polygeneration, energy storage systems, electrical vehicle, solar photovoltaics, wind energy

## Abstract

Recent advancements in sensor technologies have significantly improved the monitoring and control of various energy parameters, enabling more precise and adaptive management strategies for smart microgrids. This work presents a novel model of an energy management system (EMS) for grid-connected polygeneration microgrids that allows optimizing the management of electrical storage systems, electric vehicles, and other deferrable loads such as heat pumps. The main novelty of this model is that it incorporates both climate comfort variables and the consideration of the degradation of the energy storage capacity in the control strategy, as well as a penalty for the dumping of surpluses. The model has been applied to a smart, sustainable building as a case study. The results show that the proposed model is highly adaptable to diverse weather conditions, minimizing renewable energy losses while satisfying the energy demand and providing comfort to the building’s users. The study shows (i) that EVs’ dynamic charging schedules play a crucial role, (ii) that it is possible to minimize a battery’s degradation by optimizing its cycling, averaging one cycle per day, and (iii) the critical impact of seasonal weather patterns on microgrid energy management and the strategic role of EVs and storage systems in maintaining energy balance and efficiency.

## 1. Introduction

It is widely known that, since a couple of decades ago, we are transitioning from a fossil-based energy system to a renewable energy source (RES) one. The European Union (EU) is one of the leaders promoting this transition, as demonstrated by the enactment of several energy legislative packages. The latest, so-called “Fit for 55” package has posed three main goals: the reduction in greenhouse gas emissions of 55% by 2030 (with reference to 1990), the production of 40% of the energy demand by RESs in 2030, and the increase up to 49% in the share of RES-derived energy consumed in buildings [1]. These policies have been especially accelerated since recent international conflicts, such as the war in Ukraine, which have demonstrated the great sensibility of energy affordability of fossil resources to the developed countries [2,3]. It is not just a matter of environmental sustainability (which it still is) but has also evolved into a geo-economical strategy and dependence problem.

However, RES integration into energy systems is complex due to its intermittency and difficult predictability. Most of the current energy systems still need traditional fuel-based generators to support them when RES systems cannot satisfy the energy needs, and they strongly depend on them. Furthermore, although RES generators’ costs have decreased significantly in the last few years, making them highly competitive with respect to traditional ones, some technical problems have not yet been solved, highlighting the balance between generation and demand. Traditional bulk systems must be redesigned, and new approaches have arisen, such as introducing interconnected microgrids [4,5].

A microgrid is “a set of interconnected loads and distributed energy resources within clearly defined electrical boundaries that acts as a single controllable entity concerning the grid” [6]. Although a microgrid can involve different energy vectors (electric, heating, and cooling energy), full electrical microgrids are commonly used when using RESs, typically solar photovoltaics (PV) and wind energy, as electricity is the most efficient carrier for these technologies. Moreover, with the increase in electric vehicles in society, microgrids can host electrical vehicle (EV) charging stations (CS), which may include vehicle-to-grid (V2G) or vehicle-to-building (V2B) technologies. These systems allow EVs to behave as non-stationary electrical storage systems that can absorb the variable production of RES generators and smooth the load profile of buildings, replacing stationary battery energy storage system (ES) capacity.

Recent advancements in sensor technologies have revolutionized the monitoring and control of energy parameters within microgrids, significantly enhancing the precision and adaptability of energy management strategies [7]. These cutting-edge sensors can collect real-time data on a wide range of variables, including energy production, consumption, and storage. Then, this continuous data stream can be used to feed models for EMSs that optimize the microgrid’s performance [8].

In particular, sensors play a vital role in closely monitoring the state of battery storage systems. Sensors help predict and mitigate degradation, thus extending the lifespan of these critical components [9,10]. Furthermore, the real-time data provided by sensors also allows for dynamic adjustments to energy generation and distribution, minimizing energy losses and reducing the costs associated with renewable energy curtailment [11].

Integrating advanced sensor technologies into microgrids empowers the EMSs to implement more efficient and adaptive management strategies, ensuring a reliable and sustainable energy supply. This real-time, sensor-driven approach also supports the broader goal of transitioning to cleaner, more resilient energy systems.

Despite their advantages, microgrids have been traditionally deployed only in remote areas with limited grid access or places with weak distribution networks, due to their high initial capital costs (generators, electrical distribution, and communications infrastructure) [5]. Still, in recent years, microgrids are receiving more interest. They are being deployed in renovated urban areas to increase the aggregation of distributed generation, flexible demand, energy storage, and electric mobility. In most cases, these microgrids are behind the concepts known as Citizen Energy Communities (CECs) [12] and Renewable Energy Communities (RECs) [13].

Energy management in microgrids shows several challenges due to the increasing integration of RESs, especially solar photovoltaics and micro-wind energy. These sources are characterized by their short-term intermittency and unpredictability, making balancing generation and power demand challenging. Furthermore, as a limited number of users in the microgrid exist, oscillations and uncertainty in the energy demand are proportionally higher than in large systems. Several factors, including user behavior or weather conditions, influence demand variations. Imbalances between generation and consumption may result in dumpings of renewable energy that need to be stored or injected into the distribution network, or, on the contrary, the need to obtain energy from less sustainable sources to cover the internal demand.

Microgrids can benefit from ESs, flexible/deferrable demand, or V2G/V2B technologies that allow storage energy surpluses to be consumed later. However, the efficient management of these resources implies advanced and complex control systems that optimize the use of the storage, reduce its degradation, and guarantee the energy supply to the users [14]. The development of effective strategies for energy management that integrate these technologies and are flexible enough to adapt to the oscillations in the generation capacity, and the load is fundamental to maximizing the exploitation of RESs and guaranteeing the stability and sustainability of the microgrids.

The main objective of this study is to improve the efficiency when managing RESs in microgrids. Thus, the research has focused on developing an EMS that can optimize the operation of an electrical microgrid supported by an ES, flexible loads (such as a heat pump), and CSs for EVs with V2G capability. The system must minimize the power curtailment of the RESs and maximize its efficiency. Furthermore, focusing on smart buildings, the EMS is constrained by, on the one hand, the need to keep the maximum comfort of the building’s users, which may affect the EMS strategy drastically depending on the ambient conditions, and, on the other hand, the battery degradation when cycling (charging and discharging processes). Both aspects would require not only detailed modeling of the microgrid and its components but also of the thermal behavior of the building.

EMSs in microgrids are designed to optimize energy generation, distribution, and consumption while considering various operational constraints. Recent advancements in EMSs include the development of sophisticated optimization algorithms and control strategies. Mixed Integer Linear Programming (MILP) models are widely used for optimizing the operation and design of microgrids, as demonstrated in several studies. MILP models help minimize operational costs and emissions while considering multiple objectives such as cost efficiency and environmental sustainability. For example, in [15], a MILP model for optimizing combined heat and power (CHP) systems in an urban area with a heat distribution network is presented. It addresses capital and operating costs alongside CO_2_ emissions through a multi-objective function. Similar approaches are adopted in [16], where the authors propose a MILP model for optimizing energy management and operations planning in seaports with smart grids, considering uncertain renewable energy generation. It focuses on scheduling quay cranes, yard equipment, and ship berthing to manage energy demand and match it with supply using various pricing schemes and bidirectional trading. On the other hand, the authors of [17] incorporate reactive power and real electricity prices and evaluate the levelized cost of electricity (LCOE) and penalties in the design of their EMS.

Heuristic optimization models are usually applied when there is no possibility to linearize the model’s equations, or the objective function includes several complex objectives. That is the case, for example, of [18], in which, by applying particle swarm optimization (PSO), the method optimizes power generation and economic dispatch; or [19], which addresses the sizing of a hybrid microgrid system using multi-objective optimization algorithms, such as MOPSO (Multi-objective particle swarm optimization), PESA II (Pareto Envelope-based Selection II), and SPEA2 (Strength Pareto Evolutionary Algorithm version 2), applied to a PV/wind/diesel/battery setup.

Another significant development is the implementation of Model Predictive Control (MPC) in EMSs. MPC uses forecasts and real-time data to make optimal scheduling decisions for microgrid components, as explored in the works of [20,21]. This approach allows dynamic adjustments to energy supply and demand changes, enhancing system reliability and efficiency.

Integrating RESs such as solar photovoltaics and wind energy into microgrids poses challenges related to their intermittency and variability. Several methods were proposed for forecasting and managing RESs. Using stochastic methods and robust optimization models helps manage uncertainties in renewable generation. In [22], a robust optimization model for microgrids addressing the uncertainty of renewable energy and load demand is presented. A two-stage robust optimization balances economic efficiency and operational robustness, with the Benders dual algorithm solving the model.

On the other hand, in [23], the authors present a stochastic energy management algorithm for smart microgrids participating in the electricity market. The algorithm minimizes total costs and optimizes the sizes of various components, including wind turbines, photovoltaics, fuel cells, electrolyzers, batteries, and microturbines while addressing intermittency using the Copula method and determining market clearing prices with game theory. Another representative example is [24], which introduces a robust framework for day-ahead energy scheduling in residential microgrids with interconnected smart users, RESs, and ESs. It addresses uncertainties in RES generation and user behavior by formulating a min–max robust optimization problem to minimize energy costs while meeting constraints.

Advancements in energy storage technologies are crucial in mitigating the variability of RESs. Research by [25,26] highlights the importance of optimizing the sizing and operation of storage systems to balance supply and demand, improve voltage stability, and enhance overall system reliability. Batteries, particularly lithium-ion and sodium-nickel chloride, are widely used in microgrids due to their high energy density and efficiency. Studies such as [25,27] emphasize the need for advanced control strategies and optimization techniques to integrate batteries into microgrid operations effectively. Additionally, thermal energy storage systems and Power-to-Gas (P2G) have been explored for their potential to provide both heating and cooling, further enhancing the flexibility of microgrids [28].

On the other hand, V2G technology represents a significant advancement in integrating EVs into the energy system. V2G allows EVs to discharge stored energy back into the grid or microgrid, providing additional flexibility and support for grid operations. Research by [29,30] explores the potential of V2G technology to act as a distributed energy resource, enhancing demand response capabilities and providing ancillary services. Integrating EVs and V2G technology into microgrids offers several benefits, including improved load management, peak shaving, and enhanced resilience. Studies such as [29,30] demonstrate that V2G can help stabilize voltage levels and reduce operational costs by leveraging the distributed storage capacity of EVs.

Demand Response (DR) programs are essential for managing energy consumption and improving grid stability. DR programs incentivize consumers to adjust their energy usage in response to price signals or grid conditions. Several studies, such as [30,31,32], explore the integration of DR with microgrid management systems to achieve optimal load shifting and peak shaving. In addition to traditional DR strategies, innovative approaches such as price-based and incentive-based DR are being investigated. These methods maximize the economic benefits for consumers and utilities while reducing overall system costs and emissions.

The complexity of managing multiple Distributed Energy Resources (DERs) and various stakeholders within a microgrid has led to the exploration of decentralized and Multi-Agent Systems (MASs). These systems use distributed control and coordination strategies to enhance the efficiency and robustness of microgrid operations. For example, the study by [33] demonstrates the application of MASs for decentralized energy management, enabling local decision-making and coordination among different entities. Decentralized approaches offer advantages in terms of scalability and adaptability, especially in large and diverse microgrid networks.

The future of microgrid energy management will likely involve several key trends and innovations. Integrating advanced data analytics, machine learning, and Artificial Intelligence (AI) into EMSs will enhance the ability to predict and respond to dynamic energy supply and demand changes. Recent studies, such as [34,35], focus on this direction. In [34], the performance of various Deep Reinforcement Learning (DRL) algorithms for improving microgrid energy management is evaluated. In [35], the authors introduce a real-time dynamic optimal EMS using a DRL algorithm. The case studies presented in these works confirm these methods’ effectiveness and efficiency.

The above scientific and technical works show many approaches and models for EMSs. However, the impact of integrating efficient energy management storage with comfort parameters on the operation of the devices is still worth investigating. In particular, there can be observed a lack of studies considering the effect of the charging and discharging strategy of an ES in its degradation (which may have a significant impact on the expected capabilities and operation of the system in the medium and long term) and in the definition of a global EMS that integrates both the ES management, the CS services, and the deferrable loads, while maximizing the RES exploitation. Furthermore, very few studies have addressed the problem of minimizing the injection of RES surpluses into the electrical network. While under high electrical demand, distributed RES generation injection can be beneficial, under high rates of self-consumption or low demand periods, RES injection can increase power losses in distribution networks, create power congestion problems, and make electrical voltage control difficult [14].

This paper covers the existing knowledge gap by proposing an innovative model for an EMS that includes RES availability, thermal comfort variables, and a precise model of the cycling degradation of the energy storage devices, which completes the current state of the art. Furthermore, a penalization strategy for RES surpluses is proposed to minimize the RES curtailment or surplus injection to the electrical network, which represents an inefficacy of the system. This approach constitutes a significant innovation and a contribution to the state of the art.

A MILP approach has been adopted for the development of the EMS model. Then, a multi-objective optimization function has been defined, including the overall operation cost of the system and two penalizations: one for the batteries’ cycling and another for the RES dumping. Linear equations have defined the system operation constraints. In the case of non-linear processes, these have been approximated by linear piece-defined equations.

It has been applied to a synthetic case study to validate the model. This case study results are representative of an electrical polygeneration microgrid integrated into a smart building, and it is based on the behavior observed in the ERESMA Grid facilities (a test-bed living laboratory) at the Universidad de León (Spain), depicted in [36]. Four scenarios were considered, each representing typical conditions for each weather season of a Mediterranean country (e.g., Italy or Spain), to obtain representative results of the efficacy of the EMS strategy.

By the precise definition of the model, the EMS is expected to smartly adapt to several weather conditions, not only for the best utilization of the available RES but also to guarantee all the users’ energy and thermal comfort needs. Furthermore, the model definition can be extended to any microgrid configuration, including other energy sources or storage devices.

This work includes several innovations in the EMS definition of sustainable energy systems. It presents an innovative approach where the EMS strategy considers not only the better exploitation of the RES but also the final users’ comfort and the impact of the operation of the system in the degradation of the components, which is a critical factor for controlling the maintenance of the system and keeping the expectations on the system capability.

The remainder of this paper is organized as follows. Section 2 describes the proposed model for the EMS of a polygeneration microgrid, including a description of all the decision variables and parameters, the definition of the multi-objective function, and all the system constraints. Then, Section 3 presents the results of applying the proposed model to the case study. The system’s operation is analyzed for four representative scenarios (winter, summer, autumn, and spring), and each scenario considers four consecutive days (the considered weather forecasting horizon) evaluated in hourly intervals. Finally, the main conclusions are depicted in Section 4.

## 2. The System Model

The considered system (reported in Figure 1) is a sustainable building characterized by renewables (photovoltaics and wind turbines) that can be curtailed, an electrical storage system, a CS for V2G operations and EVs, heat pumps, an electrical connection with the main distribution grid, and a link with neighboring sustainable districts (NSD) through a network of CSs in which the EVs can be charged. In particular, the EVs can be charged from the available V2G CS in the studied microgrid and from two other different CSs: residential and public (external from the microgrid). These two latter chargers are not V2G-type chargers and have different costs. Whereas the CS in the microgrid has been considered without any charging costs, it allows the injection of electricity from the EVs into the microgrid. This is a synthetic case study representative of an electrical polygeneration microgrid where some data, such as the power generation capacity and the building electrical energy consumption, are based on those observed from a test-bed living laboratory at the Universidad de León (Spain), depicted in [36].

### 2.1. Modeling Power Production from Renewables

The produced power $\left(P_{t}^{PV,M},P_{t}^{WT,M}\right)$ is calculated based on forecasted ambient data such as solar irradiance (Equation (1)) and wind velocity (Equation (2)):(1)$$P_{t}^{PV,M}=\eta^{PV}\cdot n^{PV}\cdot A^{PV}\cdot G_{t}^{Sol}\;t=1,\ldots ,T$$
(2)$$P_{t}^{WT,M}=\left\{\begin{array}{lll} 0 & \mathrm{if}\; & s_{t}^{W}\leq s^{W,m}\\ 0 & \mathrm{if}\; & s_{t}^{W}\geq s^{W,M}\\ a\cdot (s_{t}^{W}{)}^{3}+b & \mathrm{if}\; & s^{W,m}\leq s_{t}^{W}\leq s^{W,rtd}\\ P^{WT,R} & \mathrm{if}\; & s^{W,rtd}\leq s_{t}^{W}\leq s^{W,M} \end{array}\right.\;t=1,\ldots ,T$$
where $\eta^{PV}$ is the efficiency ratio referred to each *PV* module, $A^{PV}$ is the surface of the single module, and $n^{PV}$ is the total number of modules of the PV plant. $G_{t}^{Sol}$ is the solar irradiance. The maximum power generated from the wind is a function of the wind speed $s_{t}^{W}$. If $s_{t}^{W}$ is bigger or lower than a high $s^{W,M}$ or a low $s^{W,m}$ limit value, the wind turbine will not produce power. It is important to note that power can be curtailed since the surplus power generated by RESs cannot be injected into the main grid, which means that power produced from photovoltaics $P_{t}^{PV}$ and wind $P_{t}^{WT}$ are decision variables. The curtailment mechanism is modeled through Equations (3)–(8):(3)$$P_{t}^{PV}=P_{t}^{PV,M}-P_{t}^{PV,CRT}\;t=1,\ldots ,T$$
(4)$$P^{PV,CRT,m}\leq P_{t}^{PV,CRT}\leq P_{t}^{PV,CRT,M}\;t=1,\ldots ,T$$
(5)$$P_{t}^{PV,CRT,M}=P_{t}^{PV,M}\;t=1,\ldots ,T$$
(6)$$P_{t}^{WT}=P_{t}^{WT,M}-P_{t}^{WT,CRT}\;t=1,\ldots ,T$$
(7)$$P^{WT,CRT,m}\leq P_{t}^{WT,CRT}\leq P_{t}^{WT,CRT,M}\;t=1,\ldots ,T$$
(8)$$P_{t}^{WT,CRT,M}=P_{t}^{WT,M}\;t=1,\ldots ,T$$
where
T: time horizon of the optimization problem.$P_{t}^{PV,M}$: maximum power produced by the *PV* plant [kW].$\eta^{PV}$: *PV* efficiency [%].$n^{PV}$: number of *PV* modules [units].$A^{PV}$: surface of the single *PV* module [m^2^].$G_{t}^{Sol}$: solar effective global irradiance [kWh/m^2^].$P_{t}^{WT,M}$: maximum power produced by the wind turbine [kW].$s_{t}^{W}$: wind speed [m/s].$s^{W,m}$: minimum wind speed necessary for wind turbine operations [m/s].$s^{W,M}$: maximum wind speed that allows wind turbine operations [m/s].$s^{W,rtd}$: rated wind speed [m/s].a and b: coefficients where *a* is in [kWs^3^/m^3^] and *b* is in [kW].$P^{WT,R}$: wind turbine rated power [kW].$P_{t}^{PV}$: actual power production of the *PV* plant [kW].$P_{t}^{PV,CRT}$: *PV* power curtailed, which is always positive [kW].$P_{t}^{PV,CRT,M}$: maximum *PV* power curtailed, which is always positive [kW].$P^{PV,CRT,m}$: minimum *PV* power curtailed, which is always positive [kW].$P_{t}^{WT}$: actual power production of the wind turbine [kW].$P_{t}^{WT,CRT}$: wind turbine power curtailed, which is always positive [kW].$P_{t}^{WT,CRT,M}$: maximum wind turbine power curtailed, which is always positive [kW].$P^{WT,CRT,m}$: minimum wind turbine power curtailed, which is always positive [kW].

### 2.2. Electric Vehicle Modeling

Since we are considering the microgrid optimization problem, here, the vehicle is owned by the microgrid that can be charged at three different charging infrastructures. The first is located at the microgrid, a *V2G* charging station, so that *EVs* can exchange energy with the grid bidirectionally. The second is an external *CS* outside the microgrid, and the third is a residential charging station in a different location. To ensure that, at a time t, the *EV* exists only at one specific *CS* of the locations mentioned above, three binary decision variables are introduced, indicating if the *EV* is charging at a particular site:
$x_{t}^{EV}$: 1 if the *EV* is charged at the microgrid, 0 otherwise.$x_{t}^{EXT}$: 1 if the *EV* is charged at an external site, 0 otherwise.$x_{t}^{HOME}$: 1 if the *EV* is charged at the residential area, 0 otherwise.

The energy content in the battery of the *EV* is given by
(9)$$E_{t+1}^{EV}=E_{t}^{EV}+\Delta t\left(\begin{array}{l} \left(\eta_{G2V}P_{t}^{EV,G2V}-\frac{P_{t}^{EV,V2G}}{\eta_{V2G}}\right)x_{t}^{EV}+\eta^{HOME}P_{t}^{EV,HOME}x_{t}^{HOME}\\ +\eta^{EXT}P_{t}^{EV,EXT}x_{t}^{EXT}-F^{EV}D_{t}^{EV} \end{array}\right)\;\;\;t=1,\ldots ,T$$
where
$F^{EV}$ is the electricity consumption of the *EV* in [kWh/km].$D_{t}^{EV}$ is the distance traveled by each *EV* at each time interval t.

It is important to note that the bidirectional *CS* of the microgrid cannot allow the discharging and charging of the *EV* simultaneously. Thus, the following constraints have been added:(10)$$P_{t}^{EV,G2V}P_{t}^{EV,V2G}=0\;t=1,\ldots ,T$$

Moreover, the *EV* j can be present only in one station at a time:(11)$$x_{t}^{EV}+x_{t}^{EXT}+x_{t}^{HOME}\leq 1\;t=1,\ldots ,T$$

The state of charge $SOC_{t}^{EV}$ of the *EV* can then be defined as
(12)$$SOC_{t}^{EV}=\frac{E_{t}^{EV}}{CAP^{EV}}\;t=1,\ldots ,T$$
where $CAP^{EV}$ is the charging capacity of the *EV*. In addition, bound constraints are applied as follows:(13)$$SOC^{EV,m}\leq SOC_{t}^{EV}\leq SOC^{EV,M}\;t=1,\ldots ,T$$
(14)$$0\leq P_{t}^{EV,G2V}\leq P^{EV,G2V,M}\;t=1,\ldots ,T$$
(15)$$0\leq P_{t}^{EV,V2G}\leq P^{EV,V2G,M}\;t=1,\ldots ,T$$
(16)$$0\leq P_{t}^{EV,EXT}\leq P^{EV,EXT,M}\;t=1,\ldots ,T$$
(17)$$0\leq P_{t}^{EV,HOME}\leq P^{EV,HOME,M}\;t=1,\ldots ,T$$

Equation (13) shows a bound constraint to consider the upper and lower limits of the battery of the *EV* for the three charging stations considered (microgrid, external, and residential *CSs*). The same formula can be applied to external and residential *CSs*. The variables and parameters used in the formulation above are described as follows:
$SOC^{EV,m}$: *EV* minimum state of charge [%].$SOC^{EV,M}$: *EV* maximum state of charge [%].$E_{t}^{EV}$: actual electric energy stored in the *EV* [kWh].$SOC_{t}^{EV}$: *EV* battery’s actual state of charge [%].Δt: time interval [h].$P_{t}^{EV,G2V}$: actual charging power of the facilities’ charging station [kW].$\eta_{G2V}$: charging efficiency of the facilities’ charging station [%].$\eta_{V2G}$: discharging efficiency of the facilities’ charging station [%].$P_{t}^{EV,EXT}$: actual charging power of the external charging station [kW].$\eta^{EXT}$: charging efficiency of the external charging station [%].$P_{t}^{EV,HOME}$: actual charging power of the home charging station [kW].$\eta^{HOME}$: charging efficiency of the home charging station [%].$P_{t}^{EV,V2G}$: actual discharging power of the facilities’ charging station [kW].$P_{t}^{EV,G2V,M}$: maximum charging power at the facilities’ charging station [kW].$P_{t}^{EV,V2G,M}$: maximum discharging power of the facilities’ charging station [kW].$P^{EV,G2V,m}$: minimum value of charging power of the facilities’ charging station [kW].$P^{EV,V2G,m}$: minimum value of discharging power of the facilities’ charging station [kW].$P^{EV,EXT,m}$: minimum value of charging power of the external charging station [kW].$P^{EV,HOME,m}$: minimum value of charging power of the home charging station [kW].$P^{EV,EXT,M}$: maximum allowable charging power of the external charging station [kW].$P^{EV,HOME,M}$: maximum allowable discharging power of the home’s charging station [kW].

### 2.3. Modeling the Electric Storage

The battery energy capacity $E_{t}^{ES}$ is defined through a dynamic equation (Equation (18)) and bounded between a minimum $SOC^{ES,m}$ and a maximum $SOC^{ES,M}$ (Equation (20)); the charging $P_{t}^{ES,char}$ and $P_{t}^{ES,dischar}$ powers are bounded between minimum and maximum values (Equations (21) and (22)). The battery cannot charge and discharge simultaneously; thus, the constraint Equation (23) applies:(18)$$SOC_{t+1}^{ES}=SOC_{t}^{ES}-\frac{\Delta t}{CAP^{ES}}\left(\frac{P_{t}^{ES,disch}}{\eta^{disch}}-\eta^{char}P_{t}^{ES,char}\right)\;t=1,\ldots ,T$$
(19)$$P_{t}^{ES}=P_{t}^{ES,char}-P_{t}^{ES,disch}\;t=1,\ldots ,T$$
(20)$$SOC^{ES,m}\leq SOC_{t}^{ES}\leq SOC^{ES,M}\;t=1,\ldots ,T$$
(21)$$P^{ES,char,m}\leq P_{t}^{ES,char}\leq P^{ES,char,M}\;t=1,\ldots ,T$$
(22)$$P^{ES,dischar,m}\leq P_{t}^{ES,disch}\leq P^{ES,dischar,M}\;t=1,\ldots ,T$$
(23)$$P_{t}^{ES,char}P_{t}^{ES,disch}=0\;t=1,\ldots ,T$$
where
$P^{ES,char,m}$: minimum value of charging power of the energy storage [kW].$P^{ES,char,M}$: maximum value of charging power of the energy storage [kW].$P^{ES,disch,m}$: minimum value of discharging power of the energy storage [kW].$P^{ES,disch,M}$: maximum value of discharging power of the energy storage [kW].$P_{t}^{ES,char}$: actual charging power of the energy storage [kW].$P_{t}^{ES,disch}$: actual charging power of the energy storage [kW].$E_{t}^{ES}$: energy storage stored energy [kWh].$SOC^{ES,m}$: energy storage minimum state of charge [%].$SOC^{ES,M}$: energy storage maximum state of charge [%].$CAP^{ES}$: energy storage battery capacity [kWh].$\eta^{ES}$: energy storage efficiency [%].

To minimize the battery cycles, it is necessary to define the time t at which a charging cycle starts and finishes by linking three binary variables:
$\theta_{t}$ equals to 1 if the battery is charging in time t, 0 otherwise.$\theta_{t}^{down}$ equals to 1 if a charge sequence is starting, 0 otherwise.$\theta_{t}^{up}$ equals to 1 if a charge sequence is finishing, 0 otherwise.

Similarly, it is possible to define the time t at which a discharging action starts and finishes by linking three types of binary variables:
$\beta_{t}$ equals to 1 if the battery is discharging in time t, 0 otherwise.$\beta_{t}^{down}$ equals to 1 if a charge sequence is finishing, 0 otherwise.$\beta_{t}^{up}$ equals to 1 if a charge sequence is starting, 0 otherwise.

The charging sequences are described as follows:
If, at time *t*, the storage is on charge ($\theta_{t}=1$), and at time *t −* 1, it was not on a charge sequence ($\theta_{t-1}=0$), then at time *t,* a charge sequence is starting ($\theta_{t}^{down}=1$).If, at time *t*, the storage is not on a charge sequence ($\theta_{t}=0$), and at time *t* − 1, it was on a charge sequence ($\theta_{t-1}=1$), then at time *t,* a charge sequence is finishing ($\theta_{t}^{up}=1$).If, at time *t*, the storage is on a charge sequence ($\theta_{t}=1$), and at time *t* − 1, it was on a charge sequence too ($\theta_{t-1}=1$), then there is no change in the state of charge ($\theta_{t}^{down}=0$ and $\theta_{t}^{up}=0$).If, at time *t*, the storage is on a discharge sequence ($\theta_{t}=0$), and at time *t* − 1, it was on a discharge sequence too ($\theta_{t-1}=0$), then there is no change in the state of charge ($\theta_{t}^{down}=0$ and $\theta_{t}^{up}=0$).If, at time *t*, the storage is steady ($\theta_{t}=0$) and at time *t* – 1, it was steady, too ($\theta_{t-1}=0$), then there is no change in the state of charge ($\theta_{t}^{down}=0$ and $\theta_{t}^{up}=0$).

The discharging sequences are described as follows:
If, at time *t*, the storage is on discharge ($\beta_{t}=1$), and at time *t* − 1, it was not on a discharge sequence ($\beta_{t-1}=0$), then at time *t* a discharge sequence is starting ($\beta_{t}^{up}=1$).If, at time *t*, the storage is not on a discharge sequence ($\beta_{t}=0$), and at time *t* − 1, it was on a discharge sequence ($\beta_{t-1}=1$), then at time *t*, a discharge sequence is finishing ($\beta_{t}^{down}=1$).If, at time *t*, the storage is on a discharge sequence ($\beta_{t}=1$), and at time *t* − 1, it was on a discharge sequence too ($\beta_{t-1}=1$), then there is no change in the state of charge ($\beta_{t}^{down}=0$ and $\beta_{t}^{up}=0$).If, at time *t*, the storage is on a charge sequence ($\beta_{t}=0$), and at time *t* − 1, it was on a charge sequence too ($\beta_{t-1}=1$), then there is no change in the state of discharge ($\beta_{t}^{down}=0$ and $\beta_{t}^{up}=0$).If, at time *t*, the storage is steady ($\beta_{t}=0$) and at time *t* − 1 it was steady, too ($\beta_{t-1}=1$), then there is no change in the state of discharge ($\beta_{t}^{down}=0$ and $\beta_{t}^{up}=0$).

Finally, the charging and discharging sequences are formulated as constraints for the optimization problem through Equations (28)–(31).
(24)$$\theta_{t}-\theta_{t-1}=\theta_{t}^{down}-\theta_{t}^{up}\;t=1,\ldots ,T$$
(25)$$\beta_{t}-\beta_{t-1}=\beta_{t}^{up}-\beta_{t}^{down}\;t=1,\ldots ,T$$
(26)$$\theta_{t}^{up}+\theta_{t}^{down}\leq 1\;t=1,\ldots ,T$$
(27)$$\beta_{t}^{up}+\beta_{t}^{down}\leq 1\;t=1,\ldots ,T$$
(28)$$P_{t}^{ES,char}\leq P^{ES,char,MAX}\theta_{t}\;t=1,\ldots ,T$$
(29)$$P_{t}^{ES,char}\leq P^{ES,char,MAX}\left(1-\beta_{t}\right)\;t=1,\ldots ,T$$
(30)$$P_{t}^{ES,char}\leq P^{ES,char,MAX}\left(1-\theta_{t}\right)\;t=1,\ldots ,T$$
(31)$$P_{t}^{ES,disch}\leq P^{ES,disch,MAX}\beta_{t}\;t=1,\ldots ,T$$

One of the main contributions of this work is integrating a penalty factor to minimize battery degradation. In this case, only the cycling degradation has been considered (as the calendar degradation does not depend on the operation of the battery), and it is represented in the overall objective function of the optimization problem by the term
(32)$$\sum_{t=0}^{T-1}\theta_{t}^{up}\cdot B_{c}\;t=1,\ldots ,T$$
where $B_{c}$ is the degradation cost per cycle. This virtual cost is a constant value estimated in [EUR/cycle], representing the equivalent of performance degradation cost in [EUR]. The battery replacement cost and estimated useful life (measured in cycles) can be used for calculation, depending on the battery technology. With this penalty factor, the EMS tries to minimize the use of the battery; as more cycles are used, more costs are charged.

To evaluate the battery degradation, a cycle can be determined as a function of the number of extremum ups/downs, as stated in [37]. As illustrated in Figure 2, a cycle corresponds to the period when the battery starts a charge/discharge, then switches to the opposite mode to complete a charge if the previous mode was discharging, and vice versa (to charge if it was in a discharge mode and to discharge if it was in a charge mode) [37]. For example, the number of cycles shown in Figure 2 corresponds to three cycles.

The battery is considered to be installed in an appropriate environment with constant temperature, so that ambient temperature or other factors (e.g., ramping capability or accelerated degradation by the rate of discharge) can be neglected in the degradation model of the battery. Nevertheless, if necessary, the proposed simple degradation model can be easily substituted by more detailed models.

### 2.4. Modeling the Building

The building’s thermal behavior is defined using the dynamic equation, shown in Equation (33), which models the internal temperature $T_{t}^{B}$. It is expressed as a function of the thermal fluxes exchanged between the building and the external walls $Q_{t}^{EW}$, the adjacent ceiling $Q_{t}^{AC}$, the floor $Q_{t}^{F}$, the adjacent walls $Q_{t}^{AW}$, the windows $Q_{t}^{W}$, and the fan coils $Q_{t}^{FC}$, as follows:(33)$$T_{t+1}^{B}=T_{t}^{B}+\frac{\Delta t}{C^{B}}\left(Q_{t}^{EW}+Q_{t}^{AC}+Q_{t}^{F}+Q_{t}^{AW}+Q_{t}^{W}+Q_{t}^{FC}\right)\;t=1,\ldots ,T$$

And
(34)$$Q_{t}^{EW}=\frac{T_{t}^{EW}-T_{t}^{B}}{R^{EW}}\;t=1,\ldots ,T$$
(35)$$Q_{t}^{AC}=\frac{T_{t}^{AC}-T_{t}^{B}}{R^{AC}}\;t=1,\ldots ,T$$
(36)$$Q_{t}^{F}=\frac{T_{t}^{F}-T_{t}^{B}}{R^{F}}\;t=1,\ldots ,T$$
(37)$$Q_{t}^{AW}=\frac{T_{t}^{AW}-T_{t}^{B}}{R^{AW}}\;t=1,\ldots ,T$$
(38)$$Q_{t}^{W}=\frac{T_{t}^{W}-T_{t}^{B}}{R^{W}}\;t=1,\ldots ,T$$
where
$Q_{t}^{EW}$ and $R^{EW}$: are the heat exchanged between the building and the external wall and its corresponding thermal resistance in [K/kW].$Q_{t}^{AC}$ and $R^{AC}$: are the heat exchanged between the building and adjacent ceiling and its corresponding thermal resistance in [K/kW].$Q_{t}^{F}$ and $R^{F}$: are the heat exchanged between the building and floor and its corresponding resistance in [K/kW].$Q_{t}^{AW}$ and $R^{AW}$: are the heat exchanged between the building and adjacent wall and its corresponding resistance in [K/kW].$Q_{t}^{W}$ and $R^{W}$: are the heat exchanged with the windows of the building and its corresponding resistance in [K/kW].$Q_{t}^{FC}$ and $R^{FC}$: are the thermal power the heat pump provides and its corresponding resistance in [K/kW].

$Q_{t}^{FC}$ is expressed as a function of the electric power absorbed by the heat pumps $P_{t}^{HP,EL}$ as follows:
(39)$$Q_{t}^{FC}=-P_{t}^{HP,EL}PF^{HP}\;t=1,\ldots ,T$$
where $PF^{HP}$ is the performance factor of the heat pump.

Similarly to battery degradation, a penalty factor for the thermal discomfort of the users will be incorporated into the objective function. Then, the EMS will try to minimize the consumption of the heat pump, but it will also try to keep the room’s temperature under the boundaries. The penalty for the discomfort of the users has been defined as:(40)$$\sum_{t=0}^{T-1}c^{DISC}{\left(T_{t}^{B}-T^{B,ref}\right)}^{2}\;t=1,\ldots ,T$$
where $c^{DISC}$ is a virtual cost [EUR/K^2^] that penalizes the discomfort of the users and $T^{B,ref}$ is the reference temperature [K] or setpoint to track inside the building. As can be seen, it is a quadratic function to penalize both positive and negative differences with the user’s defined setpoint. There is no reference for determining the value of $c^{DISC}$. It will depend on the relevance that the user will give to keep the comfort to the detriment of a possible improvement in the cost of operation.

### 2.5. Power Balance

The microgrid power balance is defined in Equation (41), in which the power load $P_{t}^{load}$ and the power provided by the public network $P_{t}^{GRID}$ are considered. It is expressed as follows:(41)$$P_{t}^{load}+P_{t}^{EV,G2V}+P_{t}^{ES,char}+P_{t}^{HP,EL}=P_{t}^{PV}+P_{t}^{WIND}+P_{t}^{GRID}+P_{t}^{ES,disch}+P_{t}^{EV,V2G}\;t=1,\ldots ,T$$

### 2.6. Objective Function

In the objective function expressed in Equation (42), the overall costs are minimized: *PV* and *WT* curtailment, energy bought from the grid, energy used to charge the *EV* at home, and the external *CS*. Two additional costs that correspond to the perceived discomfort and the battery degradation due to cycling are also included in the optimization as penalty factors:(42)$$J=min\sum_{t=0}^{T-1}\left[\begin{array}{c} c_{t}^{EL}P_{t}^{GRID}+c_{t}^{EV,EXT}P_{t}^{EV,EXT}+c_{t}^{EV,HOME}P_{t}^{EV,HOME}\\ +c^{PV}P_{t}^{PV,CRT}+c^{WIND}P_{t}^{WIND,CRT}+c^{DISC}{\left(T_{t}^{B}-T^{B,ref}\right)}^{2}+B_{c}\theta_{t}^{up} \end{array}\right]\cdot \Delta t$$

Subject to constraints expressed above, where
$c_{t}^{EL}$: cost of energy purchased from the grid [EUR/kWh].$c^{PV,CRT}$: cost of curtailment of the *PV* plant [EUR/kWh].$c^{WT,CRT}$: cost of curtailment of the wind turbine [EUR/kWh].$c_{t}^{EV,EXT}$: cost of energy purchased from external *CS* [EUR/kWh].$c_{t}^{EV,HOME}$: cost of energy purchased from the residential *CS* [EUR/kWh].$c^{DISC}$: cost of discomfort of the building [EUR/K].$B_{c}$: degradation cost of the battery per cycle [EUR/cycle].$T^{B,ref}$: reference temperature of the building [K].

Despite the high level of modeling of the components, this study has some limitations that must be considered. First, it does not account for the ramping capabilities of generators, charging stations for electric vehicles, energy storage systems, or heat pumps, which can impact energy management strategies. Furthermore, simple models for components, such as the battery, have been considered. More advanced models (e.g., degradation models as a function of the temperature or rate of discharge) could be used instead. Additionally, the thermal model of the building simplifies the analysis by considering a single heated space, neglecting variations between adjacent rooms. Finally, the study considers complete information in the entire time horizon, i.e., the value of the RES potential is perfectly known for all the time steps in the time horizon before running the optimization model. The estimation of the RES potential can be obtained by time-series regression models, which is out of the scope of this work. These factors suggest areas for future research to enhance model accuracy and applicability.

## 3. Results

In this section, the results of the proposed optimization algorithm are presented. The model has been implemented in Python 3.10 and used CPLEX 22.1 [38] as a solver through the PICOS library [39]. Simulations have been performed for the four seasons. Every season is represented by four consecutive days, which better illustrates the weather conditions. Moreover, this is consistent with current, accurate forecasts of renewable resources, such as solar and wind (typically, these resources can be forecasted accurately for a maximum of 4 days of hourly data based on historical data).

From a general point of view, as seen in Table 1, results show different excess/deficits in the produced renewables concerning the fixed electric demand of the microgrid. It also shows the energy fluxes in the charging station, the energy charged and discharged in the battery, and the heat pump consumption. The final column represents the Thermal Comfort Index (TCI), in degree-hours, which quantifies the deviation of indoor temperature from the desired comfort range, representing the cumulative penalty for temperatures falling below the setpoint during each season. A lower TCI indicates better performance in maintaining thermal comfort within the building.

During winter, the system experiences a significant renewable energy deficit (−250 kWh), requiring substantial energy input from the charging station (+91.74 kWh) and moderate battery discharge (+55.79 kWh) to meet demand. Spring and summer see excess renewable generation, with the battery and charging stations both storing and discharging energy as needed. Heat pump consumption is highest in spring (117.54 kWh), reflecting increased heating demand. The Thermal Comfort Index (TCI) is lowest in spring (4.40 °C·h), indicating the best thermal comfort performance, while the highest TCI in winter (19.12 °C·h) suggests greater deviation from the setpoint, reflecting colder indoor conditions.

The results of each season are given as follows.

### 3.1. Winter

The highest deficit for energy characterizes the winter season. Energy needs are estimated at 250 kWh for the four days considered, and this can be justified by the low renewable production on one side and the high demand for energy on the other, especially in cold weather. This gap in energy needs has been covered mainly by the grid, as illustrated in Figure 3.

Furthermore, heat pumps and *EV* charging are an additional energy consumption to the load, which renewable production backed by the grid has also covered. These three types of demand, i.e., load, heat pumps, and *EV* charging, have different dynamics. For instance, the load and the heat pump are fixed and must be satisfied every hour without possible shifts to different timeslots. In contrast, *EV* charging is dynamic and scheduled based on constraints related to the displacement of the *EV*, availability of the residential and external *CSs*, the charging costs, and the distance traveled. As illustrated in Figure 4, the *EV’s* battery is charged in two different charging stations: “Home”, which refers to the residential *CS*, and “external”, which refers to the external *CS*. No charging from the microgrid’s *CS* has been performed in this case. This can be explained by the constrained traveling that the *EV* has to make and the availability of the microgrid’s *CS*. In addition, the *EV* has served the grid as a storage means in particular time intervals. As can be noticed from the power balance in Figure 3 and the *EV’s* behavior in Figure 4, the *EV* has been discharged to supply electric energy to the grid thanks to the *CS’s* vehicle-to-grid capability.

An additional constraint in this scheduling problem is to satisfy the required building temperature, as illustrated in Figure 5. We can see that the reference temperature has been tracked with reasonable accuracy over the whole time horizon.

Furthermore, since the study deals with a dispatching problem, the electric storage is subject to several charging and discharging cycles, which can negatively impact its durability. Figure 6 shows the number of cycles registered for four days, considering the minimization of the number of cycles in the objective function of the optimization problem. The results show a low number of cycles, estimated at one daily cycle.

### 3.2. Autumn

Autumn has registered an energy deficit estimated at 128 kWh. This is mainly due to the low energy produced by the wind turbine. Results on the optimal power balance, the *EV’s* charging/discharging behavior, building temperature control, and the electric storage degradation are shown in Figure 7, Figure 8, Figure 9 and Figure 10, respectively.

### 3.3. Summer

During summer, a moderate excess of electric energy, estimated at 80 kWh, has been registered. In addition, the registered demand is low compared to the other seasons. Overall, different load satisfaction and building comfort requirements have been optimally satisfied, and results are depicted in Figure 11, Figure 12, Figure 13 and Figure 14, respectively.

### 3.4. Spring

The spring season has shown the highest excess of energy produced from renewables, estimated at 93 kWh. This is because of the high renewable production confronted with moderated electric demand. Figure 15, Figure 16, Figure 17 and Figure 18 show the optimal results of the power balance, the *EV* charging behavior, the temperature requirement of the building, and the state of charge of the electric storage, respectively.

Overall, the weather conditions, i.e., solar and wind, have a direct impact on the optimal scheduling of different microgrid components and the charging behavior of the *EV*. Moreover, since electric storage is the main component to ensure the dispatch of energy within the microgrid, taking into account a degradation cost, it has allowed for the minimum number of charging/discharging cycles, extending its lifetime.

## 4. Conclusions

This study introduces a novel EMS tailored for microgrids integrating polygeneration systems that address the multifaceted challenges of optimizing energy management while ensuring climate comfort and mitigating the degradation of energy storage capacity. By incorporating variables such as climate comfort, minimization of storage degradation, and a penalty for surplus generation into the scheduling strategy, our model demonstrates adaptability to diverse weather conditions.

The results observed in the case study reveal seasonal variations in energy dynamics. During winter, the highest energy deficit of 250 kWh is observed due to low renewable energy production and high energy demand, particularly for heat pumps and *EV* charging. The grid compensates for this deficit, and *EVs* contribute by discharging energy back to the grid using *V2G* capabilities. Despite the challenging conditions, building temperature requirements are maintained accurately. In autumn, a significant energy deficit of 128 kWh occurs primarily due to reduced wind turbine output. Conversely, summer sees a moderate energy surplus of 80 kWh, with lower overall energy demand. Spring registers the highest surplus of renewable energy at 93 kWh, owing to favorable weather conditions and moderate demand.

Throughout the seasons, the EMS optimally balances load satisfaction and building comfort. The *EVs’* dynamic charging schedules, influenced by travel constraints and charging station availability, play a crucial role. The battery storage system undergoes minimal cycling, averaging one cycle per day, which enhances its longevity by minimizing degradation. This study underscores the critical impact of seasonal weather patterns on microgrid energy management and the strategic role of *EVs* and storage systems in maintaining energy balance and efficiency.

The results of this study demonstrate that the proposed model provides significant advantages in adapting to diverse weather conditions, effectively minimizing renewable energy losses while meeting energy demand and ensuring user comfort. Notably, this study shows that including comfort variables and degradation and curtailment penalties may affect the operation strategy of an EMS, improving its results, which constitutes a significant advance in the state of the art.

Future work will focus on integrating detailed modeling of the components as constraints in the scheduling optimization problem for more accurate results and the validation of the model with field tests.

## Figures and Tables

**Figure 1 sensors-24-07122-f001:**
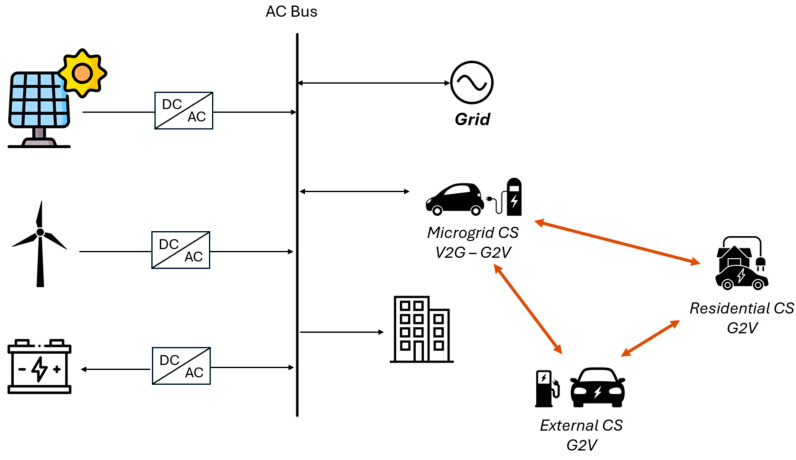
Simplified scheme of the studied microgrid.

**Figure 2 sensors-24-07122-f002:**
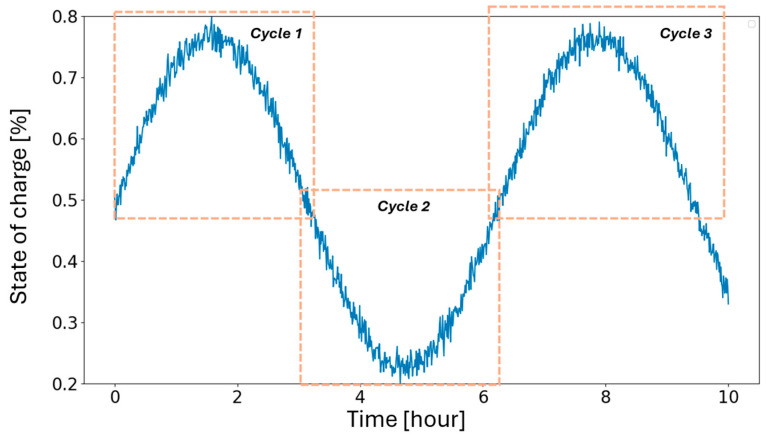
Hypothetical example to illustrate how cycles are counted in an electric storage.

**Figure 3 sensors-24-07122-f003:**
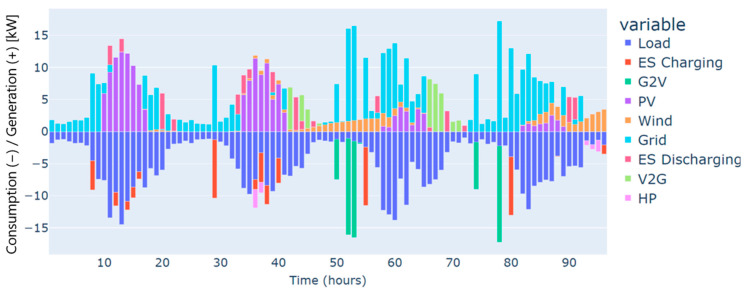
The power balance of the microgrid for a typical four-day period in winter.

**Figure 4 sensors-24-07122-f004:**
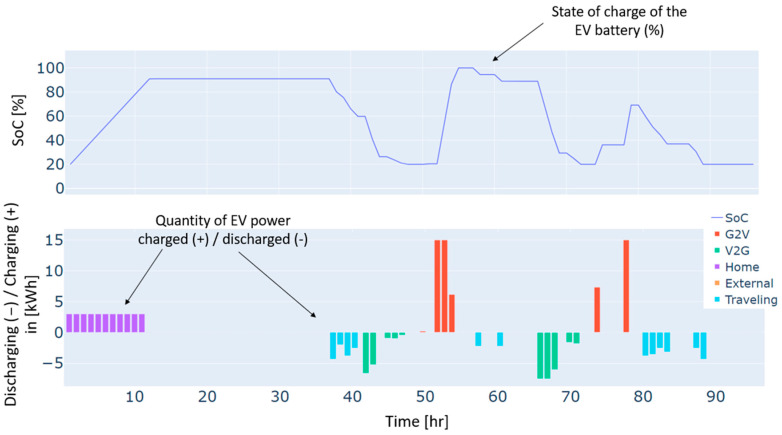
The behavior of the EV for a typical four-day period in winter.

**Figure 5 sensors-24-07122-f005:**
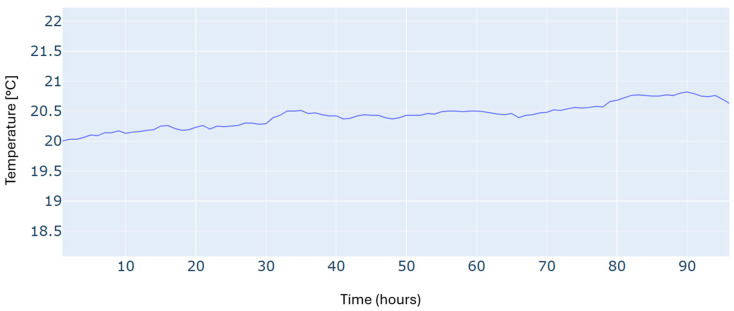
Results of the building temperature for a typical four-day period in winter.

**Figure 6 sensors-24-07122-f006:**
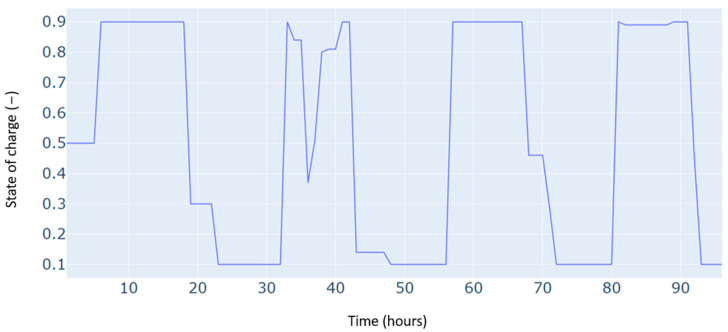
The state of charge for the microgrid storage for a typical four-day period in winter.

**Figure 7 sensors-24-07122-f007:**
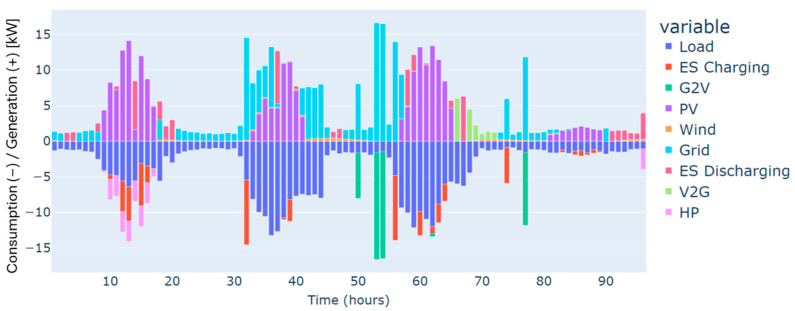
The power balance of the microgrid for a typical four-day period in autumn.

**Figure 8 sensors-24-07122-f008:**
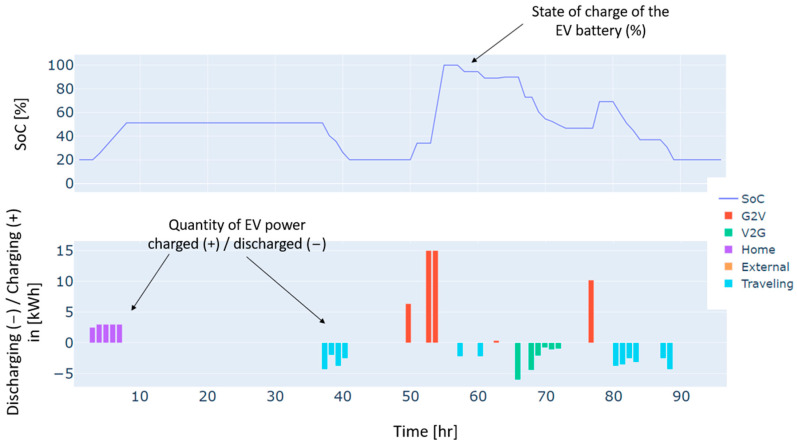
The behavior of the EV for a typical four-day period in autumn.

**Figure 9 sensors-24-07122-f009:**
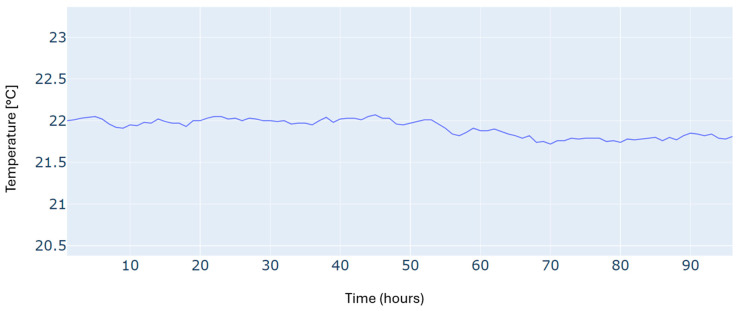
Building temperature for a typical four-day period in autumn.

**Figure 10 sensors-24-07122-f010:**
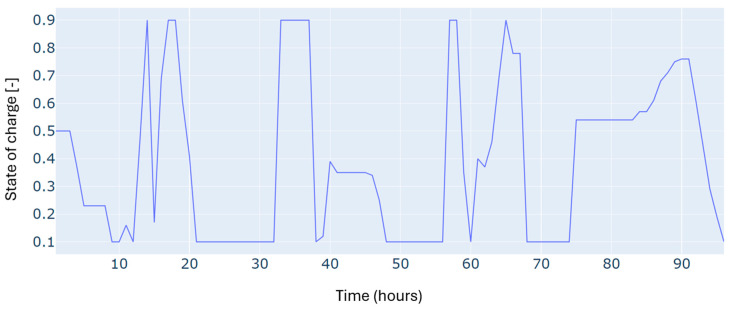
The state of charge of the microgrid storage for a typical four-day period in autumn.

**Figure 11 sensors-24-07122-f011:**
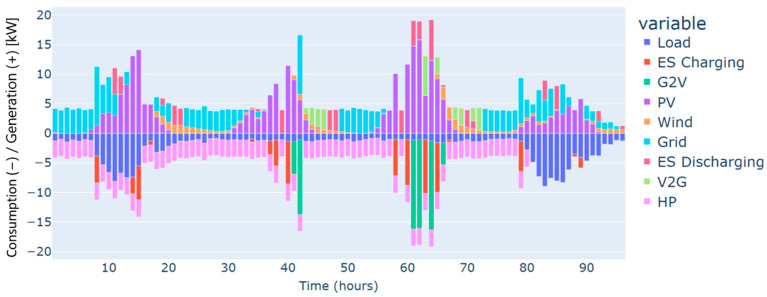
The power balance of the microgrid for a typical four-day period in summer.

**Figure 12 sensors-24-07122-f012:**
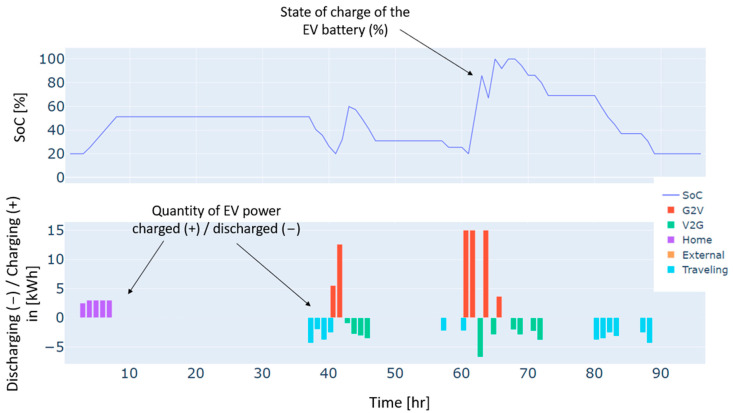
The behavior of the EV for a typical four-day period in summer.

**Figure 13 sensors-24-07122-f013:**
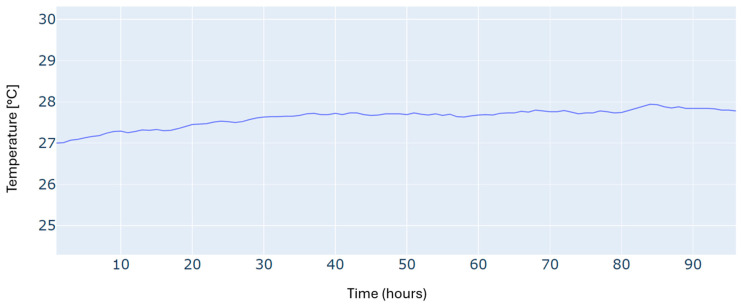
Building temperature for a typical four-day period in summer.

**Figure 14 sensors-24-07122-f014:**
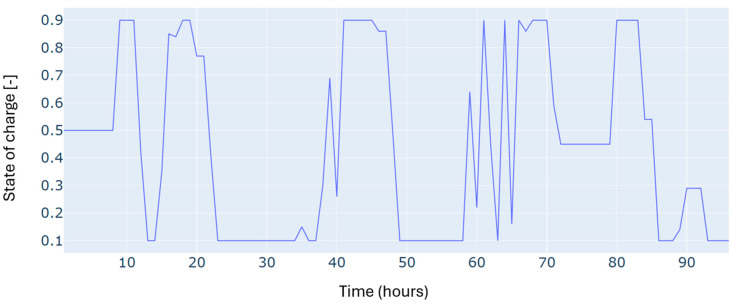
The microgrid storage state of charge is for a typical four-day period in summer.

**Figure 15 sensors-24-07122-f015:**
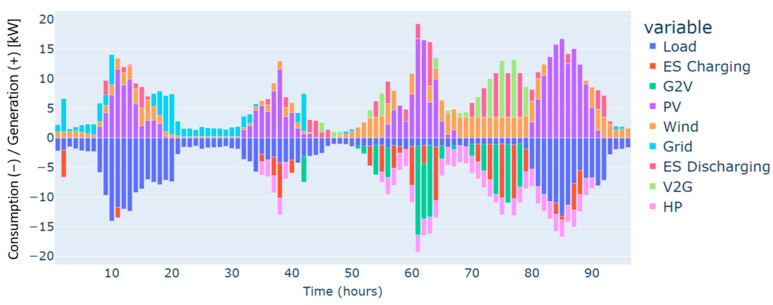
The power balance of the microgrid for a typical four-day period in spring.

**Figure 16 sensors-24-07122-f016:**
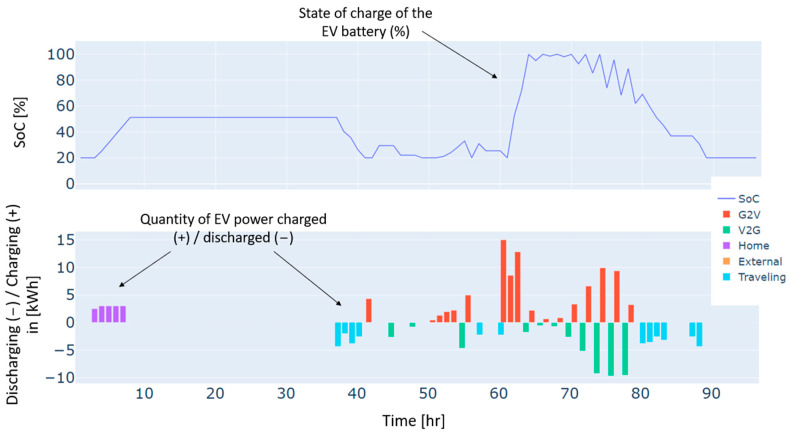
The behavior of the EV for a typical four-day period in spring.

**Figure 17 sensors-24-07122-f017:**
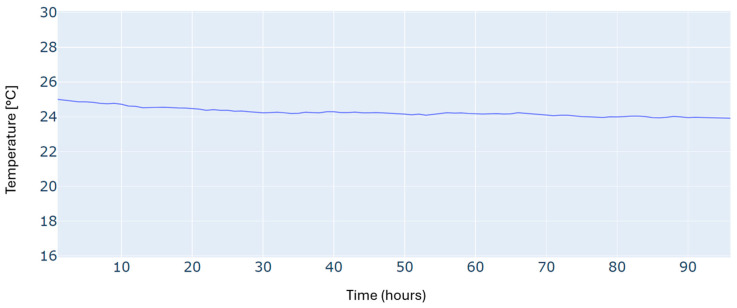
Building temperature for a typical four-day period in spring.

**Figure 18 sensors-24-07122-f018:**
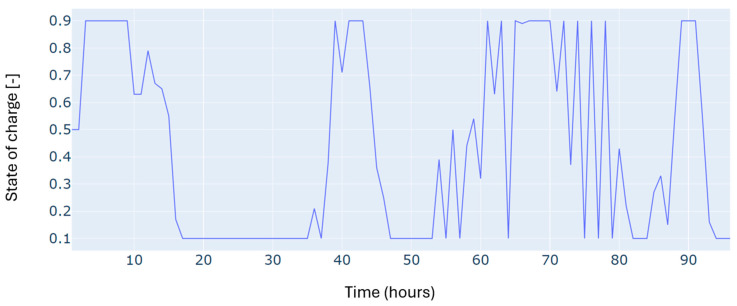
The state of charge for the microgrid storage for a typical four-day period in spring.

**Table 1 sensors-24-07122-t001:** Total renewable production and demand (4-day periods).

	Solar PV [kWh]	Wind [kWh]	Load [kWh]	RES Excess/Deficits [kWh]	CS Charge/Discharge [kWh]	Battery Charge/Discharge [kWh]	Heat Pump [kWh]	TCI [deg.h]
Winter	162	58	470	−250	+91.74 −38.34	+55.79 −45.58	4.52	19.12
Spring	307	156	370	+93	+89.86 −37.21	+84.94 −76.06	117.54	4.40
Summer	264	39	223	+80	+71.28 −22.82	+62.31 −55.38	86.82	5.91
Autumn	220	11	359	−128	+61.44 −15.20	+56.57 −50.85	21.54	8.74

## Data Availability

The data presented in this study are available on request from the corresponding author due to restrictions on privacy and confidentiality, which prevent open sharing of the dataset.

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
