# Peer review of "Energy Management System for Polygeneration Microgrids, Including Battery Degradation and Curtailment Costs"

_sensors, 2024, doi:10.3390/s24227122_

Round 1

Reviewer 1 Report

Comments and Suggestions for Authors

The paper presents the development of a microgrid optimization process that integrates different types of renewable energy sources (RES), such as solar and wind. It is undoubtedly an interesting paper that deals with current topics such as energy efficiency and user comfort. 

The authors are suggested to clarify some points in the paper. Firstly, elaborate on the description of the battery degradation model. Although its inclusion is mentioned, the details about how factors such as depth of discharge, charge/discharge rate, and temperature are integrated and how the degradation is translated into an economic cost still need to be clarified. 

A more detailed explanation is required to quantify the "cost of discomfort" associated with the deviation of the building temperature from the reference temperature. How is this cost determined? What level of deviation is considered tolerable? This will allow a better understanding of the influence of thermal comfort on the proposed Energy Management System (EMS). 

In order to bring the model closer to reality, incorporating uncertainty in the RES generation and demand is recommended. Although the variable nature of these parameters is recognized, the current model does not explicitly consider uncertainty in its calculations. What can be said about this issue? 

It is recommended that the authors provide detailed information on the origin of the data used in the simulations. Are the data used to test the model completely hypothetical, or do some come from real measurements at the ERESMA Grid facilities?

Author Response

Comment 1: The authors are suggested to clarify some points in the paper. Firstly, elaborate on the description of the battery degradation model. Although its inclusion is mentioned, the details about how factors such as depth of discharge, charge/discharge rate, and temperature are integrated and how the degradation is translated into an economic cost still need to be clarified.

Response 1: According to the reviewer's comment, the description of the battery degradation penalty factor has been extended. It is a simplified model that measures the number of cycles of the battery (considering a cycle as a function of the number of extremum ups/downs of charging/discharging) and applies a virtual cost that can be determined as the rate of the replacement cost and the useful lifespan of the battery measured in cycles. The battery is considered to be installed in an appropriate environment with constant temperature and, as stated in the limitations paragraph (added at the end of section II after this revision), no ramping capabilities, rates of discharge, temperature, or other factors have been considered in the degradation model of the battery. This issue is worth mentioning, and the authors will consider it in further research. This study aims to analyze the impact of a penalty factor for the degradation of the battery in the EMS. The proposed model is open to include more accurate models for this and other aspects.

Comment 2: A more detailed explanation is required to quantify the "cost of discomfort" associated with the deviation of the building temperature from the reference temperature. How is this cost determined? What level of deviation is considered tolerable? This will allow a better understanding of the influence of thermal comfort on the proposed Energy Management System (EMS).

Response 2: We completely agree with the reviewer's comment. Thus, a more detailed description of the discomfort penalty factor has been included after this manuscript revision. As stated in the text, there is no reference to define the value of this factor. It will depend on the relevance that the user will give to keep the comfort to the detriment of a possible improvement in the cost of operation. In future research, the model's sensibility to the value of this penalty factor will be investigated.

Comment 3: In order to bring the model closer to reality, incorporating uncertainty in the RES generation and demand is recommended. Although the variable nature of these parameters is recognized, the current model does not explicitly consider uncertainty in its calculations. What can be said about this issue?

Response 3: We completely agree with the reviewer's comment. RES generation is characterized by high uncertainty, significantly affecting the EMS behavior. The proposed model considers complete information in all the time horizon, i.e., the value of the RES potential is perfectly known for all the time steps in the time horizon before running the optimization model. The estimation of the RES potential can be obtained by time-series regression models, which is out of the scope of this work. Nevertheless, we will consider the effect of prediction models for the RES estimation in the EMS performance in future research. This aspect has been included also as a limitation of the study. 

Comment 4: It is recommended that the authors provide detailed information on the origin of the data used in the simulations. Are the data used to test the model completely hypothetical, or do some come from real measurements at the ERESMA Grid facilities?

Response 4: In the introduction section of the manuscript, it has been explained that the proposed model has been applied to a synthetic case study, which is representative of an electrical polygeneration microgrid integrated into a smart building. It is based on the behavior observed in the ERESMA Grid facilities (a test-bed living laboratory) at the Universidad de León (Spain). Data such as power generation capacity and energy consumption come from actual measurements taken at the test facilities. Other data are synthetic, such as the power consumption of the EVs. In this state of research, the proposed EMS has only been simulated and not tested in actual operation conditions, which we expect to do soon in future research. These aspects have been clarified again in Section II, where the system model is presented.

Finally, we thank the reviewer for their thorough and detailed review of our work, which undoubtedly contributed to improving the paper's initially submitted version. We hope that the new version meets the reviewer's quality standards, and we remain at your disposal for further comments or revisions of the paper.

Reviewer 2 Report

Comments and Suggestions for Authors

The manuscript presents a comprehensive study on an energy management system (EMS) designed for microgrids that integrate polygeneration systems. It addresses challenges related to optimizing energy management while ensuring climate comfort and mitigating the degradation of energy storage capacity. The findings highlight seasonal variations in energy dynamics, with specific insights into how the system adapts to different weather conditions to maintain energy balance and efficiency. Overall, the manuscript is well-written and could be published with a few minor revisions. Here are some suggested revisions:

1. Some sentences are complex and may hinder readability. Simplifying or breaking down longer sentences will enhance clarity, allowing readers to better grasp key findings.

2. Ensure that terms such as "electric vehicle (EV)" and "renewable energy sources (RESs)" are defined upon first use and consistently applied throughout the manuscript, including uniform formatting of acronyms like "CS" and "BESS."

3. To improve the clarity of the discussion on seasonal energy dynamics, it is advisable to present key quantitative data in a tabular format. This approach will allow for a more straightforward comparison of energy deficits and surpluses across different seasons, thereby enhancing the manuscript’s overall clarity.

Author Response

Comment 1: Some sentences are complex and may hinder readability. Simplifying or breaking down longer sentences will enhance clarity, allowing readers to better grasp key findings.

Response 1: We thank the reviewer for their comment. The whole manuscript has been revised, and some paragraphs have been rewritten to increase clarity.

Comment 2: Ensure that terms such as "electric vehicle (EV)" and "renewable energy sources (RESs)" are defined upon first use and consistently applied throughout the manuscript, including uniform formatting of acronyms like "CS" and "BESS."

Response 2: All acronyms in the manuscript have been revised according to the reviewer's suggestion.

Comment 3: To improve the clarity of the discussion on seasonal energy dynamics, it is advisable to present key quantitative data in a tabular format. This approach will allow for a more straightforward comparison of energy deficits and surpluses across different seasons, thereby enhancing the manuscript's overall clarity.

Response 3: Table 1 in the Results section has been enhanced, including the total microgrid's CS charging and discharging energy, the heat pump consumption, and the Thermal Comfort Index (TCI) of each scenario, which helps to understand the obtained results.

Finally, we thank the reviewer for his thorough and detailed review of our work, which undoubtedly improved the paper's initially submitted version. We hope that the new version meets the reviewer's quality standards, and we remain at your disposal for further comments or revisions of the paper.

Reviewer 3 Report

Comments and Suggestions for Authors

·       Required more detailed modeling of the components, such as the thermal behavior of buildings and the specific characteristics of different energy storage systems. This would enhance the accuracy of the EMS optimization.

·       If possible, Include the real-world data from existing microgrids could validate the proposed model to evaluate the practicality of your proposed model

Improve the quality of the images 

Comments on the Quality of English Language

Minor editing required 

Author Response

Comment 1: Required more detailed modeling of the components, such as the thermal behavior of buildings and the specific characteristics of different energy storage systems. This would enhance the accuracy of the EMS optimization.

Response 1: As suggested, the description of both the thermal model of the building and the energy storage system has been revised. The associated penalty factors with the thermal discomfort of the users and the battery cycling degradation have been explained in more detail. We hope that the new description will be more straightforward for future readers.

Comment 2: If possible, Include the real-world data from existing microgrids could validate the proposed model to evaluate the practicality of your proposed model.

Response 2: In the introduction section of the manuscript, it has been explained that the proposed model has been applied to a synthetic case study, which is representative of an electrical polygeneration microgrid integrated into a smart building. It is based on the behavior observed in the ERESMA Grid facilities (a test-bed living laboratory) at the Universidad de León (Spain). Data such as power generation capacity and energy consumption come from actual measurements taken at the test facilities. Other data are synthetic, such as the power consumption of the EVs. In this state of research, the proposed EMS has only been simulated and not tested in actual operation conditions, which we expect to do soon in future research. These aspects have been clarified again in Section II, where the system model is presented.

Comment 3: Improve the quality of the images.

Response 3: We do not clearly understand the aspects of figure quality the reviewer refers to (resolution, labeling, etc.). Nevertheless, the quality of all figures in the manuscript has been reviewed, and we hope they will look appropriate in the final published version of the paper.

Finally, we thank the reviewer for his thorough and detailed review of our work, which undoubtedly improved the paper's initially submitted version. We hope that the new version meets the reviewer's quality standards, and we remain at your disposal for further comments or revisions of the paper.

Reviewer 4 Report

Comments and Suggestions for Authors

The following issues should be addressed to improve the quality of this paper:

a. In the abstract section, it is recommended that the authors include a sentence summarizing the conclusions.

b. Once an abbreviation has been defined, it does not have to be further defined when it is used again. EMS should not be defined several times.

c. Subsections 3.1, 3.2 … 3.6 need to be renumbered because are included in section 2. Also, subsections form section 3 should be renumbered.

d. Make sure the equations are appropriately cited in the text.

e. No information is given about the software used for solving the optimization problem.

f. Discussion of the results needs to be improved.

g. The importance and benefits of this research should be better underlined.

h. The authors should define abbreviations before using them (line 136 – MOPSO, PESA II, SPEA2, …).

i. It is recommended that the authors indicate the limitations of this study.

j. Impersonal writing should be used consistently throughout the manuscript.

Author Response

Comment 1: In the abstract section, it is recommended that the authors include a sentence summarizing the conclusions.

Response 1: The abstract has been revised, and the final sentence summarizing conclusions has been rewritten according to the reviewer's observation.

Comment 2: Once an abbreviation has been defined, it does not have to be further defined when it is used again. EMS should not be defined several times.

Response 2: We thank the reviewer for their comment. The whole text has been revised, and duplicate definitions have been removed.

Comment 3: Subsections 3.1, 3.2 … 3.6 need to be renumbered because are included in section 2. Also, subsections form section 3 should be renumbered.

Response 3: We thank the reviewer for their observation; the number of sections and subsections has been fixed.

Comment 4: Make sure the equations are appropriately cited in the text.

Response 4: The whole text has been revised considering this aspect.

Comment 5: No information is given about the software used for solving the optimization problem.

Response 5: The results section clarifies now that the model was implemented in Python 3.10 and used CPLEX as a solver through the PICOS library.

Comment 6: Discussion of the results needs to be improved.

Response 6: The results section has been revised and enhanced. Furthermore, new information (charged and discharged energy in the CS and the cumulated degrees hour from the set point) has been included in Table 1, which summarizes the key results, and results have been analyzed in depth.

Comment 7: The importance and benefits of this research should be better underlined.

Response 7: We completely agree with the reviewer, and Section IV (Conclusion) has added a new paragraph highlighting the benefits and key findings of the conducted research.

Comment 8: The authors should define abbreviations before using them (line 136 – MOPSO, PESA II, SPEA2, …).

Response 8: The definitions of all abbreviations have been included now.

Comment 9: It is recommended that the authors indicate the limitations of this study.

Response 9: We completely agree with the reviewer, and a new paragraph indicating the study's limitations is included at the end of the second section, which presents the proposed model.

Comment 10: Impersonal writing should be used consistently throughout the manuscript.

Response 10: The whole text has been revised considering this aspect.

Finally, we thank the reviewer for his thorough and detailed review of our work, which undoubtedly improved the paper's initially submitted version. We hope that the new version meets the reviewer's quality standards, and we remain at your disposal for further comments or revisions of the paper.

Round 2

Reviewer 1 Report

Comments and Suggestions for Authors

Modifications and additions made to the paper are accepted accordingly.